# Distribution of Dental Fluorosis in the Southern Zone of Ecuador: An Epidemiological Study

**DOI:** 10.3390/dj11030071

**Published:** 2023-03-03

**Authors:** Eleonor María Vélez-León, Alberto Albaladejo-Martínez, Paulina Ortíz-Ortega, Katherine Cuenca-León, Ana Armas-Vega, María Melo

**Affiliations:** 1Department of Surgery, Faculty of Medicine, University of Salamanca, 37007 Salamanca, Spain; 2School of Dentistry, Catholic University of Cuenca, Cuenca 010107, Ecuador; 3Latin American Network of Research on Fluorides and Dental Fluorosis, Cartagena 130009, Colombia; 4Private Consultation, Cuenca 010107, Ecuador; 5School of Dentistry, Hemisferios University, Quito 170527, Ecuador; 6Faculty of Medicine and Dentistry, Department of Stomatology, University of Valencia, 46010 Valencia, Spain

**Keywords:** dental fluorosis, oral health, fluoridation, prevalence, Ecuador

## Abstract

In recent decades, the increase in fluoride exposure has raised the numbers of dental fluorosis in fluoridated and non-fluoridated communities In Ecuador, but the last national epidemiological study on DF was conducted more than a decade ago. The objective of this cross-sectional descriptive study was to determine the prevalence, distribution and severity of dental fluorosis (DF) using the Dean index in 1606 schoolchildren aged 6 to 12 years from urban and rural environments in provinces that make up the Southern Region of Ecuador. Participants met the inclusion criteria which were age, locality, informed consent document and no legal impediment. The results are presented using percentage frequency measures and chi-square associations. The prevalence of dental fluorosis was 50.1% in the areas of Azuay, Cañar and Morona Santiago, with no significant differences (x^2^ = 5.83, *p* = 0.054). The types of DF found most frequently were very mild and mild in all provinces; a moderate degree was more prevalent in Cañar (17%). There was no significant association (*p* > 0.05) between sex and the presence of dental fluorosis and, with respect to severity, the most frequent degree was moderate at the age of 12 years. The prevalence of dental fluorosis in the area evaluated is high, especially in the light and very light degrees, with a tendency toward moderate levels. It is necessary to carry out studies on the factors that are predisposing to the development of this pathology in the population studied. This research is an update regarding this pathology in Ecuador, so it is concluded that it is necessary to continue developing studies based on the findings obtained, thus contributing to the public health of the country.

## 1. Introduction

The use of fluoride (F) to promote oral health has always involved a balance between protection against caries and the risk of fluorosis [1]. Dental fluorosis is a condition of enamel development caused by fluoride intake during the period of dental development, with the critical time in its development being between six and nine months of age for primary dentition [2,3]. For permanent dentition, the period varies depending on the type of tooth and the duration of exposure to F during amelogenesis in the first three years of life [2,3,4,5]. Clinically, this pathology can be observed as white opacities in mild cases to more severe black and brown discolorations or pitting of the enamel [6]. Differential diagnoses include, and are not limited to, early carious lesions, molar hypo-mineralization, developmental disorders of enamel and dentin including amelogenesis imperfecta, Turner’s hypoplasia, tetracycline staining and dental manifestations of celiac disease [6,7,8,9].

In mild and moderate cases of DF, the affected dentition is resistant to caries [10], while severe lesions with increased enamel porosity show increased fragility on the external surface, which can easily fracture with mechanical chewing forces [11], becoming more severe and affecting the overall appearance of the teeth [6,7,8,10,11]. Structural damage may increase in the long term depending on the severity [7]. Although the occurrence of minor lesions is generally socially accepted, moderate and severe forms can sometimes compromise esthetics and generate treatment needs in individuals and concerns about the impact on quality of life [8,12,13,14].

Dental fluorosis (DF) is the most common adverse effect of fluoride use in the prevention of dental caries [6,7]. However, the reduction in dental caries has been accompanied by an increase in the prevalence of DF, ranging from 7.7% to 80.7% in areas where fluoridated water is available and from 2.9% to 42% in areas without fluoridated water [8]. Its importance in public health lies in the fact that DF is a population biomarker of fluoride exposure in young children; recommendations on the use of fluoride should be based on evidence of a risk–benefit tradeoff between a preventive benefit against dental caries and a risk of fluorosis [4,10]. For this reason, research on various aspects of dental fluorosis has gained momentum in several countries [7,11,12], as it is also part of oral health policies. The formulation and implementation of these public health policies may vary from one country to another. They depend on multiple factors, including the constant interaction between national and international public health agencies and scientific knowledge, i.e., the evidence or information available from reliable sources, of national or foreign origin [13].

The main known source of F is water [13,15]. From 1945, when community water fluoridation began in the USA, to 2015, 25 countries have practiced this measure [11,16] for its known role as a key strategy in the prevention of dental caries. However, water is not the only source; F is also present in fish, tea, formula milk, salt, and fluoride supplements (toothpaste, fluoride varnish application, and mouthwash) [17]. Against this background, in recent decades, exposure to F in the pediatric population in different periods of their lives has increased, resulting in higher prevalence rates of DF in fluoridated and nonfluorinated communities [18]. The most appropriate period to provide F while maintaining a balance between the risk and benefit is still unknown [19]. However, controversy remains about the variety of F sources and the presence of risk factors such as climate, altitude, and geographical conditions [20]. Based on the above, it is thought that, in regions with altitudes higher than 1500 m, the renal filtration and clearance of some substances is less efficient, so that an increase in the concentration of fluoride then concentrates in teeth and bones [18,21,22].

In Ecuador, community measures for caries prevention have been in place since 1974 through the Water Fluoridation Program [23], based on WHO/PAHO [24] recommendations. However, when inconveniences were detected, such as the lack of drinking water service in communities, especially in rural areas, a few years later, this measure lost its effect.

In 1986, through chemical monitoring of the natural fluoride content in the country’s drinking water [25], it was concluded that only the northern part of the country contained high levels of fluoride, while in the rest of the country fluoride was very low or non-existent. In this perspective, in 1997 the National Program for Fluoridation of Salt for Human Consumption was implemented, since it is a product that is marketed without problems and at low cost, which accessible to the entire population and establishing itself as the main systemic fluoridation measure in the country for the prevention of tooth decay. The concentration of fluoride in salt is established nationally at a concentration of 200 to 250 ppm [26] for communities where the concentration of fluoride does not exceed 0.7 ppm in drinking water; otherwise, it is recommended to opt for the consumption of fluoride-free salt.

As regards the control values of the prevalence of DF in the population, it is necessary to mention that no recent epidemiological studies have been carried out in Ecuador at a national level, but there are data from more than a decade ago, which report fluorosis values of 4% in non-fluoridated areas water and 93% in the northern areas of the country where water is naturally fluoridated. In this framework, one of the measures taken by the Ministry of Public Health and the Directorate of Stomatology of Ecuador was to recommend periodic dental fluorosis studies and monitoring of the amount of fluoride in water, since the change in sociodemographic conditions in certain regions of the country due to accelerated urbanization in both urban and rural areas would be accompanied by new and multiple water sources per community [25]. It should be noted that the southern part of the country, where this study is located, has reported low fluoride levels ranging from 0 to 0.11 ppm [27]. This is a region characterized by a high prevalence of caries [28] and few studies of DF [29], although in other regions of the country with non-fluoridated water, factors that could be related to the development of this pathology have already been identified, such as the continuous intake of fluoride supplements in pre-school children [30].

In this context, the objective of this study is to follow up this pathology (DF) in terms of prevalence and severity in school-age children of both sexes in urban and rural settings in a region that does not report updated epidemiological data.

## 2. Materials and Methods

### 2.1. Design

This is an observational, relational, cross-sectional study on the prevalence and distribution of DF severity in the Southern Region of Ecuador, 2019.This research received the approval of the Board of Directors of the Health and Welfare Unit under code No. 048CD-2019 (approved on 14 February 2019) of the Catholic University of Cuenca-Ecuador. The researchers obtained informed consent from the legal representatives of participants.

The prevalence of dental fluorosis included the categories of absent and present, while the degree of fluorosis included normal, questionable, very mild, mild, moderate and severe.

### 2.2. Sample

According to the report of the National Institute of Statistics and Census (INEC) (2010 report) [31], the estimated population of children aged 6 to 12 years was 183,081. A one-stage random cluster sampling was performed, the invitation was made to all children who belonged to 36 schools which were part of this research and the sample size was calculated by convenience through the EPIDAT 4.0 program, resulting in 1938 participants with a confidence rate of 95% and a margin of error of 0.5%. The inclusion criteria towere that the participants presented an informed consent form signed by their legal representative, that they had permanent teeth, that they belonged to the study localities, that they were between 6 and 12 years of age and that they did not present any physical or legal impediment to their examination. The final sample consisted of 1606 school-age children from the provinces of Azuay, Cañar and Morona Santiago, of whom 826 were boys and 780 girls.

### 2.3. Inclusion Criteria

Among the inclusion criteria, it was taken into account that the children reside in Azuay, Cañar and Morona Santiago, that they have permanent dentition, that they are within the contemplated ages, that their representatives had signed the informed consent and finally that they did not present any physical or legal impediment to be part of the study.

### 2.4. Exclusion Criteria

The exclusion criteria were those who did not meet the required age, did not have permanent dentition, did not wish to participate, were not in the study locality and had physical or legal impediments. Children who were within this framework were not part of this study.

### 2.5. Calibration

The training and calibration process was performed by certified dental professionals. The diagnostic criteria used were those of the Dean index, included in the oral health surveys stipulated by the WHO [32]. The calibration process consisted of five theoretical and three practical sessions; clinical images of dental fluorosis were used and identification was also performed on extracted teeth. For reliability and reproducibility of the examiners, 2 groups of 10 children (10 with DF and 10 without pathology) from local schools were clinically examined on two different days and one week apart. The Kappa statistical value reached by the 6 calibrated professionals was 0.88, which shows an adequate level of agreement.

### 2.6. Examination

To be included in the study, prior to the examination the schoolchild had to have an informed consent signed by a guardian and a previously completed questionnaire that included demographic information such as name, age, sex, social number, and place of residence. All examinations were performed in natural light using a mouth mirror and following standard infection control guidelines [33]. Under Dean’s criteria all permanent teeth present in the mouth were examined. Those teeth with 50% of the clinical crown erupted were included for the examination; the degree of severity of DF was determined by the most affected teeth [34]

The selected criteria corresponded to:

0 = Healthy dental organs, enamel smooth, bright, usually creamy white.

1 = Doubtful, when the enamel showed slight alterations in enamel translucency, which could be white spots or scattered dots.

2 = Very slight, when there were small white or opaque spots similar to paper scattered on the dental crown and affecting less than 25% of the tooth surface.

3 = Mild, when there were streaks or lines across the tooth surface, and the white opacity affected between 25% and 50% of the tooth surface.

4 = Moderate, when the enamel showed marked involvement with brown staining.

5 = Severe, when the enamel surface was severely affected, and hypoplasia manifested as excavated areas with intense brown staining and a corroded appearance.

#### Statistical Analysis

The analysis began with the general analysis of the prevalence of DF and then showed the levels of alteration. The results are expressed through percentage frequency measures. In addition to establishing the association between variables, the chi-square statistic was used. The statistical programs IBM^®^ SPSS v.27 (New York, NY, USA) and JASP^®^ 0.16.2 (Amsterdam, The Netherlands) were used. The analysis was performed in SPSS V27, and the significance level was 5% (*p* < 0.05).

## 3. Results

A total of 1606 school children aged 6 to 12 years from urban and rural settings in the provinces of Azuay (573), Cañar (610) and Morona Santiago (423) were examined. All participants had to meet inclusion criteria: age, locality, signed informed consent and no legal impediments.

With regard to the overall prevalence of dental enamel, 50.1% of the schoolchildren presented dental enamel, with no significant differences between Azuay, Cañar and Morona Santiago (x^2^ = 5.83, *p* = 0.054). (Figure 1).

In terms of the prevalence and distribution of DF levels by urban/rural environment and by province, a statistical association was found within the three regions (Azuay, Cañar and Morona Santiago) (*p* = 0.002). In Azuay and Morona Santiago, the most frequent severity of presentation of fluorosis was very slight with a presence of 20.2% and 17.3%, respectively.

In Cañar, the levels of very slight to moderate were at about the same frequency (approximately 17% in each type). A significant association (*p* < 0.05) was identified between the severity of fluorosis and province, with the absence of the intense type in Cañar and the presence of the moderate type in the same province, much higher than in the other groups (Table 1).

The severity of fluorosis found in each of the environments according to the provinces revealed significant differences (*p* = 0.002) only in the province of Azuay. The analysis found that the presence of moderate fluorosis was significantly higher in urban in comparison with rural areas, in Azuay specifically (Table 2).

No relationship was found between the sex of the schoolchildren (*p* > 0.05) and the presence of dental fluorosis; however, age was related to the level of alteration (x^2^ = 103.6; *p* = <0.001), the main differences being that, in children 7 and 8 years old, the moderate level was significantly lower than the very light and light levels. In addition, in children 12 years old the very light level was more prevalent than the light level, and the moderate level was much higher than the light and very light levels (Table 3).

## 4. Discussion

This study, whose objective was to investigate the prevalence of DF and its severity, is the first carried out in the southern region of the country, in provinces with different climatic and altitude characteristics such as Azuay, Cañar and Morona Santiago, which are apparently non-fluoridated areas, so the data obtained can serve as a starting point for future research. The results show a high prevalence of DF in the three provinces evaluated, with no significant differences between them (*p* > 0.05), coinciding with previous studies carried out in some Latin American cities [31,35,36,37,38,39] and in Ecuador [30,40,41,42] (Figure 2). The fluoride incorporated into the organism from water sources, processed or not processed food, or by accidental ingestion, through the consumption of toothpaste, constitutes the main source for pathologies at the dental enamel level [43], with an effect variable according to the period of contact of the mineral with the tooth in its formation stage [44].

The areas evaluated belong to the Sierra region of Ecuador, whose altitude levels vary between 900 and 3300 m above sea level. Although isolated studies of F in water have been reported [27,30], in an attempt to comply with the recommendation of the MSP in 1996 that monitoring should be annual, this process has not been standardized, resulting in a lack of knowledge of the exact concentrations of fluoride in the water.

The evaluation of the presence of pathology requires a careful protocol and the use of one of the two systems or indices to measure the severity of the pathology, the Dean index [36] or the Thylstrup and Fejeskov index (tfi) [45], whose use is validated and considered reliable in the analysis of DF, with slight differences between them limited to the analysis protocol [46,47]. For that reason, the use of the Dean index was required in this study, because it was considered comparable to previous studies carried out in similar Ecuadorian populations, which evidenced the high presence of mild fluorosis in the evaluated population [40], related to the consumption of food, water, tooth brushing, pharmacies’ use [30], triggering enamel defect [48] and demineralization processes associated with dental cavities when the DF increase [49].

The roughness of the affected enamel surfaces due to the consistent effect of DF constitutes a site of accumulation of bacterial plaque and food deposit, causing the development of carious processes [50] that, according to previous studies referenced and executed in Ecuador, has shown a frequent presence both in decidual dentition and in definitive teeth [28]. In thus grave state, an intervention of restorative treatments is required [51], creating a higher cost for the intervention at dental clinics, and there is a shared tendency towards a constant restoration circle, mainly because poor development related to the buccal cavity is generally associated with a lack of proper hygienic habits performed by the patient and a low level of health knowledge in their carers [52]. In addition to this fact, the aesthetic alterations that the presence of the pathology produces trigger a decrease in the quality of life of the individuals who suffer [53].

Although the presence of fluorosis has been associated in previous studies in similar regions [54] to the accidental ingestion of fluorinated elements in daily use [30], the level of fluorosis did not show a significant difference in terms of the provinces evaluated, which may be explained by the equally comparable geographical situation, climate, soil, lithology, and topography of the populations that have been evaluated, mainly dedicated to mining, which can explain the presence of fluorosis [55].

When urban and rural areas were considered, only in the province of Azuay was there a difference in the degree of DF, where the moderate degree was significantly higher in the urban area than in the rural area, a result that agrees with a previous study [49]. This can be explained by the different life conditions of the local residents from both areas, where inhabitants living in urban areas are privileged according to their socioeconomic level, with proper access to health services and thus a larger contact to extra sources of fluoride compared to rural residents [41].

One of the important limitations of the study was the cross-sectional evaluation carried out, which prevents elements of comparison that establish a relation with other variables that, at a certain moment, could explain the cause of the pathology. However, considering age group leads to questioning the preventive strategies established by the country’s health entities, so several evaluations for future studies require to be executed in a periodic manner with the same populations.

Therefore, it is necessary to carry out new follow-up studies that address the influence of other variables as causal elements of the pathology, which were not considered in this study but would help to better understand the results. The studies carried out in Ecuador do not follow an established analysis protocol, which makes it difficult to obtain comparative data [53], and therefore difficult to gain parameters for analysis.

As clinicians, the high prevalence of patients with fluorosis leads us to reflect on the need to implement measures for the diagnosis and early detection of the presence of the pathology, seeking to act vis strategies for the individual and the community for control, while understanding that the sooner the execution the better, as preventive processes and rehabilitation can obtain much better results. Management and detection of these injuries require to be executed, before they combine with other elements such as dental plaque leading to surface damage that can produce enamel retention areas and causing irreversible loss of dental structure. The high presence of DF in mild degrees leads us to reevaluate the procedures and hygienic therapies that can be executed on the population, necessitating an imminent incorporation of educational processes for the inhabitants around hygienic techniques and tooth brushing, and the future professional dentists can recognize these oral health factors and give them the importance they deserve.

## 5. Conclusions

It is necessary to mention that no recent national epidemiological studies have been carried out in Ecuador, but there are data from more than a decade ago, which report fluorosis values of 4% in non-fluoridated areas.

Therefore, the present investigation represents updated information for the public health of our environment and, among the results, in the population evaluated there are high values of prevalence of dental fluorosis, with a predominance of “mild” and “very mild” levels.

It can be concluded that the sociodemographic and cultural conditions since the last study in this region have changed, so new research is recommended to verify these new findings and thus develop research that will allow us to analyze the factors associated with the pathology presented.

## Figures and Tables

**Figure 1 dentistry-11-00071-f001:**
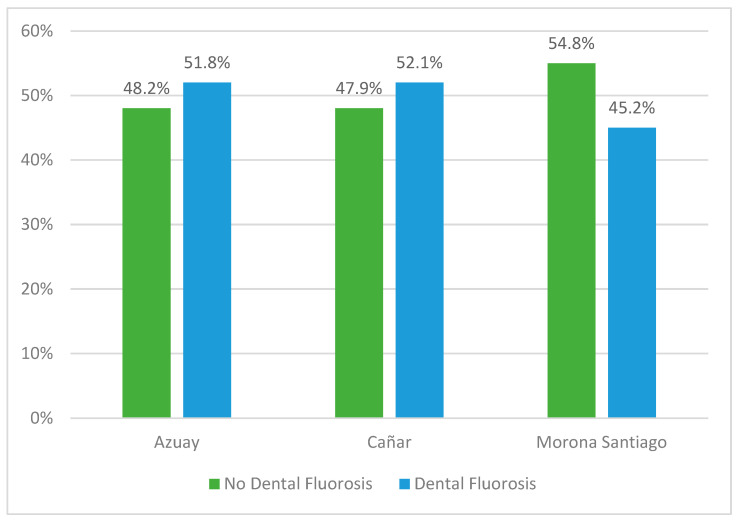
Prevalence of Dental Fluorosis by province.

**Figure 2 dentistry-11-00071-f002:**
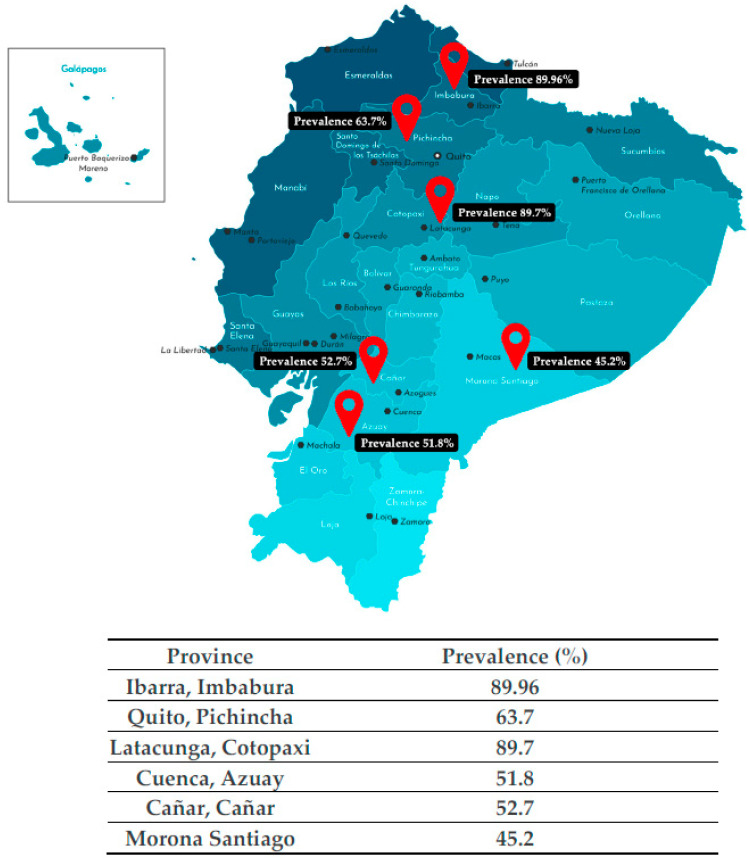
Geographical distribution of the areas studied and comparison with other studies. Source: Image taken from iStock. https://www.istockphoto.com/es/vector/ilustraci%C3%B3n-aislada-vectorial-mapa-administrativo-simplificado-de-ecuador-en-gm1217481314-355406824, accessed on 20 October 2022.

**Table 1 dentistry-11-00071-t001:** Distribution of the Severity of DF by Province.

Fluorosis	Azuay		Cañar		Morona Santiago		Total	
	n	%	n	%	n	%	n	%
Normal	276	48.2	292	47.9	232	54.8	800	49.8
Debatable	2	0.3	0	0.0	0	0.0	2	0.1
Very slight	116	20.2	105	17.2	73	17.3	294	18.3
Slight	113	19.7	108	17.7	64	15.1	285	17.7
Moderate	61	10.6	104	17.0	51	12.1	216	13.4
Intense	5	0.9	0	0.0	3	0.7	8	0.5
Excluded	0	0.0	1	0.2	0	0.0	1	0.1

**Table 2 dentistry-11-00071-t002:** Prevalence and distribution of DF levels by urban and rural setting.

Fluorosis	Azuay				Cañar				Morona Santiago			
	Urban		Rural		Urban		Rural		Urban		Rural	
	n	%	n	%	N	%	n	%	n	%	n	%
Normal	168	44.2	108	37.6	110	39.9	182	38.1	128	46.5	104	43.0
Debatable			2	0.7								
Very slight	66	17.4	50	17.4	40	14.5	65	13.6	44	16.0	29	12.0
Slight	63	16.6	50	17.4	36	13.0	72	15.1	34	12.4	30	12.4
Moderate	45	11.8	16	5.6	35	12.7	69	14.4	27	9.8	24	9.9
Intense	3	0.8	2	0.7					2	0.7	1	0.4
Excluded							1	0.2				
x^2^	26.400				1.866				6.300			
*p*	<0.001				0.087				0.275			

Note: x^2^: 34.576; *p*: 0.002.

**Table 3 dentistry-11-00071-t003:** Distribution and relationship of DF with age and sex.

			Debatable	Very Light	Light	Moderate	Intense	Exclude	x^2^ (*p*)
Sex	Men	n	0	144	134	125	4	1	9.37 (0.095)
		%	0.0	35.3	32.8	30.6	1.0	0.2	
	Woman	n	2	150	151	91	4	0	
		%	0.5	37.7	37.9	22.9	1.0	0.0	
Age	6	n	0	11	8	2	0	0	103.6 (0.000 **)
		%	0.0	52.4	38.1	9.5	0.0	0.0	
	7	n	1	41	58	14	0	0	
		%	0.9	36.0	50.9	12.3	0.0	0.0	
	8	n	0	58	55	22	2	0	
		%	0.0	42.3	40.1	16.1	1.5	0.0	
	9	n	1	50	47	32	3	1	
		%	0.7	37.3	35.1	23.9	2.2	0.7	
	10	n	0	38	43	42	0	0	
		%	0.0	30.9	35.0	34.1	0.0	0.0	
	11	n	0	47	50	25	2	0	
		%	0.0	37.9	40.3	20.2	1.6	0.0	
	12	n	0	49	24	79	1	0	
		%	0.0	32.0	15.7	51.6	0.7	0.0	

(**) = (*p* < 0.01).

## Data Availability

https://ucacueedu-my.sharepoint.com/:u:/g/personal/mvelezl_ucacue_edu_ec/EWCY_ysABVdJl2VgLAcHtMEBqioojuJMGPZoxFaGIaJUAA?e=BCxPGl (accessed on 18 December 2021).

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
