# Peer review of "Distribution of Dental Fluorosis in the Southern Zone of Ecuador: An Epidemiological Study"

_dentistry, 2023, doi:10.3390/dj11030071_

Round 1

Reviewer 1 Report

Dear Authors, 

Congratulations on the work you have done and presented in this manuscript. I believe that your article is really interesting and can be of interest for the general reader. I suggest to revise the manuscript, please see the attachment.

Author Response

Estimado revisor,

Consulte el archivo adjunto.

Reviewer 2 Report

Insufficient information:

The study's objectives were to report the condition of dental fluorosis (DF) in Ecuador and compare the difference in urban and rural areas. However, this needed to be clearly stated at the end of the Introduction. Besides, more information on the fluoride policy (concentration of fluoride in water and salt? ) is needed. The author reported the data according to three cities separately. However, the background information on these three provinces needs to be more comprehensive (concentration of fluoride in water and salt? any difference?). The author compared the prevalence of DF in urban and rural areas in three provinces separately and explored its relationships with participants' sex and current age only. No investigation on the relationship between the prevalence of DF and fluoride exposure was conducted, which is more important. It is a pity that the study did not collect information on fluoride exposure.

Study design:

Is the sampling process random? How many schools were selected and invited to participate in the study? Were all children in the participating schools invited? When conducting the dental examination, which teeth were recorded? All fully erupted teeth? More information should be provided. How many examiners were involved in the study? Only 10 children were recorded twice for intra- and inter-examiner reliability during the calibration, which is not enough. At least 5-10% of participants should be re-examined throughout the study to monitor the intra- and inter-examiner reliability.

Statistical analysis and errors in reported data:

The statistical methods stated for the data analysis need to be more specific. 

In the Results, the way reported could be more logical as some reported data were inconsistent with the figure. For example, "51.8% with no significant differences (p>0.05)" (line 142); according to Figure 1, 51.8% is the prevalence of DF in participants from the city of Azuay. So what does "no significant difference" mean? It is also stated that "a significant difference was found....between Canar and Santiago (P=0.27), between Azuay and Morona (P=0.37)..." (line 143), as the P values were >0.05 and there should be no significant difference.

In the following paragraph, "a significant difference was identified between the types of fluorosis and province" (line 154), which should be stated as "a significant association was identified....". Please use the severity of fluorosis to replace types of fluorosis. Also, the use of "level of alteration" is confusing.

"The type of fluorosis found in each of the environments according to the provinces revealed significant differences (P=0.002) only in the province of Azuay" (line 160). The interpretation is incorrect as the P = 0.002 was an overall test applied to all three regions, not just for Azuay. On the other hand, "the presence of moderate fluorosis was found to be significantly higher in the urban areas and rural areas (Table 2)", which only refers to Azuay, not to the other two provinces.

In Table 3, no decimals are to be used for the age. Also, the interpretation of the chi-square test is inappropriate.

Discussion:

The discussion provides little interpretation of the results. It just states more background and basic information. Not so much helpful inspiration can be derived from the discussion. 

Conclusion:

The author stated, "The population evaluated in the study region registered......" but in the table in Figure 2, these three provinces were unregistered areas with fluoride content in water. What is the meaning of unregistered? Does the fluoride content mean fluoride concentration? 

Overall, the quality of the paper is low. Professional English editing is much needed. Substantial revisions are required to present data and analysis and discuss based on the findings. The reporting should follow the STROBE guidelines.

Author Response

Dear reviewer,

Reviewer 3 Report

The manuscript was well written and easy to follow the research question and methods. The findings will add to evidence with impact more in the regions studied. However, the generalizability is limited because of the differences in different geographical regions.

Line 44: The reference #11, "NADELMAN, P.; MAGNO, M.B.; PITHON, M.M.; CASTRO, A.C.R. de; MAIA, L.C. Does the Premature Loss of 269 Primary Anterior Teeth Cause Morphological, Functional and Psychosocial Consequences? Braz Oral Res 2021, 35, 270 doi:10.1590/1807-3107BOR-2021.VOL35.0092", does not support the sentence. This should be replaced/removed/ a new sentence will be needed to cite that paper.

Author Response

Dear reviewer,

Round 2

Reviewer 2 Report

-        The abstract has not been revised to reflect the revisions in the main text, especially the result, which is critical.

-        DF and FD (10 times) have been used as the abbreviation of dental fluorosis inconsistently, and dental fluorosis has been used from time to time after the abbreviation has been used.

-        Line 183, the results should be written as “Regarding the overall prevalence of DF, 50.1% of the schoolchildren presented DF with no significant differences among Azuay, Canar and Morona Santiago (x2=(please insert the chi-square statistic), p=(please insert the overall p-value), Figure 1)”. As the overall p-value was bigger than 0.05, there is no need to perform the pairwise comparison, thus the results “When a pairwise comparison… (Figure 1)” can be deleted.

-        The “Note” at the bottom of Table 2 can be deleted, please change the p-value “0.000” to “<0.001”.

-        Line 214, please revise “age was related to the level of DF severity…”

-        Please change “p=0.000” to “p<0.001” (Line 215)

-        Table 3, please change the p-value “0.000” to “<0.001”.

-        Further English editing is needed.
